# A Feasibility Study of the Usefulness of the TEMPS-A Scale in Assessing Affective Temperament in Athletes

**DOI:** 10.3390/medicina58020195

**Published:** 2022-01-27

**Authors:** Katarzyna Białczyk, Jan Kłopocki, Jacek Kryś, Maciej Jaskulski, Anna Lewandowska, Robert Szafkowski, Karol Ogurkowski, Derek Pheby, Karl Morten, Marcin Jaracz

**Affiliations:** 1Health Economy Division, Collegium Medicum in Bydgoszcz, Nicolaus Copernicus University in Torun, 85-092 Bydgoszcz, Poland; jacek.krys@gmail.com (J.K.); m.jaskulski@jurasza.pl (M.J.); 2Clinical Neuropsychology Division, Collegium Medicum in Bydgoszcz, Nicolaus Copernicus University in Torun, 85-092 Bydgoszcz, Poland; janklopocki@gmail.com (J.K.); szafkowskir@gmail.com (R.S.); marcinjaracz@cm.umk.pl (M.J.); 3Rehabilitation Division, Collegium Medicum in Bydgoszcz, Nicolaus Copernicus University in Torun, 85-092 Bydgoszcz, Poland; a.lewandowska@cm.umk.pl (A.L.); ogurkowski@gmail.com (K.O.); 4Society and Health, Buckinghamshire New University, High Wycombe HP11 2JZ, UK; derekpheby@btinternet.com; 5Nuffield Department of Women’s & Reproductive Health, University of Oxford, Oxford OX3 9DU, UK; karl.morten@wrh.ox.ac.uk

**Keywords:** affective temperament, athletes, sport attitude

## Abstract

*Background and objectives*: Current studies show an important role of affective temperament in sport performance. The aim of this study was to assess the feasibility of the use of the TEMPS-A scale, by using it to examine five dimensions of affective temperament in three groups of athletes. We hypothesized that temperament may be a predisposing factor to the level of commitment and type of training. *Materials and methods*: The study group (N:71, 33 female) consisted of professional canoeists (N:25, aged 18–30), sports pilots (N:21, aged 19–57) and non-professionals regularly performing aerobic exercises (N:25, aged 23–33). The Affective Temperament of Pisa, Paris and San Diego Autoquestionnaire (TEMPS-A) was used to evaluate affective temperament dimensions. Statistical analysis was performed using non-parametric tests. *Results*: The TEMPS_A scale shows good internal consistency; a hyperthymic temperament was associated with different factors compared to other temperament traits. The most prevalent trait in the study group was hyperthymic temperament. The study group scored higher on hyperthymic and lower on depressive and anxious temperaments when compared with the general population. Canoeists scored higher on cyclothymic temperament compared with non-professional athletes and on cyclothymic and irritable dimensions in comparison with pilots. Pilots obtained significantly lower scores on irritable and anxious temperaments than non-professional athletes. Females scored higher on both hyperthymic and irritable dimensions. No significant differences were found in respect of depressive, cyclothymic and anxious traits. Age was negatively correlated with cyclothymic and irritable temperament scores. *Conclusions*: TEMPS-A scale is a useful tool for assessing affective temperament in athletes. The results suggest that affective temperament may be a factor influencing physical activity engagement. Different types of activities may be connected with different temperament dimensions. Younger athletes present a higher tendency to mood lability and sensitivity to environmental factors. However, further research is needed, involving larger numbers of subjects.

## 1. Introduction

Physical activity may involve different levels of performance. Nowadays, many people take on physical activity as a leisure activity that is intended to improve mood and self-esteem, and to help cope with stress [1,2]. Studies show that approximately 70% of adults take on regular, moderate physical exercise that meets international public health guidelines [3,4]. Some of them get more involved in training and commit nearly as much time and effort as professional athletes. Temperament is biologically determined and relatively stable. It is present from childhood, and is the basis for the formation of later human personality traits.

The role of temperament in sport performance has been studied before. Current research shows that early temperament features may influence the formation of later physical activity habits, especially in men. Boys with higher activity and intensity of pleasure engaged in greater amounts of adolescent physical activity [3]. Furthermore, a longitudinal association of temperament traits, negative affect and physical activity with gender specificity in young children has been found [5]. Regular exercise may itself promote the development of some behavioral features. Additionally, different temperament features may be associated with different types of exercise (aerobic, anaerobic, technical). It is also evident that athletes differ from the general population in terms of temperament traits. They are more dominant and prone to experience pleasure, moreover they show higher levels of harm-avoidance [6,7]. The affective temperament theory was developed by an American researcher—Hagop Akiskal [8,9]. Affective temperament is a genetically determined construct which consists of five dimensions or traits, viz., depressive, cyclothymic, hyperthymic, irritable, and anxious [10]. The concept of affective temperament is derived from the works of E. Kraepelin, who believed that the affective temperaments are widely distributed in the general population and play an important role in social adaptation. In their extreme intensity, they are, however, related to the elevated risk of affective disorders [10,11].

A growing body of research has been dedicated to the identification of the functional adaptive attributes of affective temperaments, and their role in adaptation to stressful, demanding situations, as well as in professional functioning. Different dimensions of affective temperament are associated with varying patterns of emotional response to stressful situations [12]. Depressive, cyclothymic, irritable and anxious temperaments have been related to poorer stress coping and poorer performance in stressful situations as well as to higher vulnerability to psychiatric, especially affective, disorders [8,9,13,14]. On the other hand, the hyperthymic temperament is associated with an improved ability to perform tasks in difficult conditions, higher creativity and initiative and better decision-making ability in stressful situations [11,15]. In spite of the growing body of research on the functional qualities of affective temperaments, we find a scarcity of research on their role in sports and athletic performance.

The aim of this study was to evaluate the five dimensions of affective temperaments in athletes—professional canoeists, sports pilots and non-professionals—regularly performing aerobic exercises, and to assess the usefulness of the Affective Temperament of Pisa, Paris and San Diego Autoquestionnaire (TEMPS-A) instrument in making this evaluation. We hypothesize that hyperthymic temperament may be related to a higher level of involvement in physical activity. Likewise, the hyperthymic temperament may be more frequent in professional athletes, as competitive sport at the professional level requires much self-discipline, devotion and resilience, all of which are characteristics of the hyperthymic temperament. Consequently, we hypothesized that depressive, cyclothymic, irritable and anxious temperaments would be less represented in professional athletes, as their qualities are disadvantageous in the context of competitive endeavors. To our knowledge, this is the first study to directly compare affective temperaments in professional and non-professional athletes.

## 2. Materials and Methods

### 2.1. Subjects

A total of 100 self-reported sets of questionnaires, including the TEMPS-A scale, a survey for obtaining demographic information (age, gender) and sports performance were used. Ultimately, 71 sets of questionnaires were returned, by 38 male and 33 female participants. These subjects were aged 18–57 years, with a mean age of 27.11 ± 7.73. All the subjects were healthy, without a history of serious neurological, psychiatric or somatic diseases.

### 2.2. Methods

Affective temperaments were evaluated by TEMPS-A [2]. This is a 110-item self-report autoquestionnaire for assessing five dimensions: depressive, cyclothymic, irritable, hyperthymic and anxious. The Polish version of TEMPS-A was used [16]. Subjects were asked to answer YES or NO for each question. Dimensions of affective temperament are associated with specific behavioral features, e.g., a depressive temperament is connected with a tendency to pessimism, worry, guilt-proneness and difficulties in experiencing joy, as well as a preoccupation with inadequacy, failure and negative events. A cyclothymic temperament is characteristic of persons presenting unstable mood, labile self-esteem and overconfidence alternating with low self-confidence. A hyperthymic temperament characterizes people with high openness for novel experiences, high activity and leadership, and also with higher levels of risk-behaviors, stress resistance and grandiosity. The irritable dimension may manifest itself in dysphoric, explosive reactions and restlessness. The anxious temperament is associated with a higher risk of anxiety disorders and defense reactions [9].

In the sample from this study, TEMPS-A showed good reliability, with Cronbach-alpha coefficients of 0.70 for depressive, 0.81 for cyclothymic, 0.82 for hyperthymic, 0.71 for irritable, and 0.86 for anxious temperament.

### 2.3. Statistical Analysis

The Shapiro–Wilk test was applied to assess the normality of distribution of variables. Because of the nonparametric character of variable distributions, the Mann–Whitney U-test was used to evaluate between-group differences. For more than two group differences, the Kruskal–Wallis ANOVA test was used. Correlations between variables were assessed using Spearman’s rho coefficient. The internal consistency of TEMPS-A was evaluated using the Kuder–Richardson 20 test.

## 3. Results

The subjects were divided into three groups: (1) 25 professional canoeists, aged 18–30 years with a mean age of 22.12 ± 2.92; (2) 25 non-professionals regularly performing aerobic exercises aged 23–33 years, mean age 27.36 + 2.50; and (3) 21 sports pilots aged 19–57 years, mean age 32.76 ± 11.25.

### 3.1. Internal Consistency of TEMPS-A

The first step of analysis involved determining the internal consistency of TEMPS-A subscales (Table 1), and factor analysis of the main components (Table 2, Figure 1). This procedure was required to check the usefulness of the TEMPS-A questionnaire in evaluating affective temperament in the study group.

As is shown in Table 1, in the study group, the anxious temperament shows significant positive correlations with the depressive (r = 0.49), cyclothymic (r = 0.47) and irritable (r = 0.44) traits. Irritable dimensions positively correlate with cyclothymic ones (r = 0.54). No significant associations between the hyperthymic temperament and the remaining dimensions were observed. Factor analysis shows a significant association of cyclothymic, irritable and anxious temperaments with factor I and of hyperthymic temperament with factor II.

### 3.2. The Affective Temperament Profile

Table 3 presents the TEMPS-A scale results in the study group and the difference between males and females. Gender difference in two dimensions, hyperthymic (*p <* 0.04) and irritable (*p <* 0.04), were observed. Females scored higher on both the hyperthymic and irritable dimensions. No significant differences on depressive, cyclothymic and anxious temperament were found.

The scores on five dimensions of affective temperament obtained by subjects from three groups of athletes and the significance of differences between groups are shown in Table 4. Three groups of athletes differentiated cyclothymic and irritable temperaments. According to this observation, a comparison of the results between individual groups of athletes was made. Canoeists scored more highly on cyclothymic temperament compared to non-professional athletes (*p* < 0.05), and on cyclothymic (*p* < 0.01) and irritable (*p* < 0.01) dimensions in comparison with pilots. Pilots obtained significantly lower scores on irritable (*p* < 0.03) and anxious (*p* < 0.05) temperament than non-professional athletes.

The differences between groups are confirmed by multidimensional analysis (Table 5 and Figure 2). Correlation analysis shows negative associations between age and scores on two dimensions of affective temperament: cyclothymic (R = −0.48, *p* < 0.01) and irritable (R = −0.27, *p* < 0.01). Younger persons showed a significantly higher score on the cyclothymic temperament, indicating greater mood lability, and a higher score on irritable temperament associated with sensitivity to environmental factors and a tendency to explosive reactions.

## 4. Discussion

Contemporary health policy promotes the beneficial health effects of physical activity. This results in an increase in interest in amateur sport in the general population. Currently, interest in the psychological factors determining sports achievement has also significantly increased. Many authors have pointed out that individual factors such as personality and temperament traits are the basis for the development of the skills needed in competitive sport [10,17,18]. Additionally, supporting mental health and well-being, especially in young athletes, is very important [19]. Affective temperament studies can be very useful in this regard. In this study, the TEMPS-A autoquestionaire was applied for the evaluation of affective temperament in professional and non-professional athletes. The statistical analysis shows good internal consistency and internal correlation of TEMPS-A subscales, which indicate the usefulness of this scale for the measurement of affective temperament in the study group. In comparison with an earlier study, hyperthymic temperament is a different dimension compared with the remaining affective temperament traits, probably connected with different neurobiological mechanisms [11,16,20,21].

In this study, subjects with higher levels of physical activity compared to the general population were involved. Differences were also found between affective temperament scores in the entire study group in comparison with the general population. This may suggest that the type of dominant temperament may influence the degree of engagement in sports. The Polish validation of the TEMPS-A scale performed in the general population shows that the hyperthymic type was less common (0.50 ± 0.19 vs. 0.63 + 0.15) whereas depressive and anxious where more common (0.20 + 0.10 vs. 0.32 ± 0.15 and 0.16 + 0.15 vs. 0.30 ± 0.21 respectively) than in the study group [16]. Moreover, investigated females presented a different pattern of temperament profile in comparison with the general populations described in various studies. The results of many studies show higher rates of depressive, cyclothymic and anxious temperament in females, and higher rates of hyperthymic temperament in males in different populations [15,16,22]. Our study was performed on athletes, whose activity requires a higher focus on goals and who exhibited a greater prevalence of hyperthymic and irritable temperament traits in women than in men. This may be related to the specific psychological predispositions of athletes, especially women, who are successful in competitive sport. These results correspond with observations in bank managers, paramedics and athletes. These professions are associated with high stimulation, high task orientation behavior and activity under psychological pressure, which require efficient operation under stress [11,15]. Our study may also be consistent with results that indicate greater rates of bipolarity and impulsivity in extreme or high-risk athletes [23]. It is likely that these traits may be associated with a higher probability of risky and/or maladaptive behavior in a stressful situation. The findings of Scandinavian authors show a greater risk of stress-related disorders, including burnout, in highly competitive junior athletes. The results show an association between fear of failure and burnout and the level of psychological stress in athletes, related to the individual-oriented dimensions of fear of failure [24].

The main finding in this research was the identification of differences in the affective temperament profile in athletes representing different sports activities. Contrary to our hypotheses, we did not find expected differences in temperament between non-professional and professional athletes. Non-professional athletes, as well as pilots, scored lower in cyclothymic temperament compared to canoeists. This may be partially related to the fact that canoeists were the youngest group from the whole study sample, as the level of cyclothymic temperament was related to age in our correlational analyses. Interestingly, pilots show lower rates of irritable, anxious and cyclothymic temperaments in comparison to other groups studied. This may be connected with specific psychological abilities in accordance with the requirements of their sport discipline, and also with longer training time and the shaping of special psychological skills, especially since the pilots group involved in this study was older than the canoeists group. Our data confirmed earlier results obtained by an Italian researcher indicating higher scores on hyperthymic and lower scores on depressive, cyclothymic or irritable temperaments in applicants taking medical exams and aptitude tests. A hyperthymic temperament was an important factor in choosing a profession, and also in passing entrance examinations [25,26].

We also found a significant correlation between age and affective temperament scores. Younger age was associated with higher level of cyclothymic and irritable traits. This may indicate that younger athletes present a stronger tendency to mood lability, and probably higher sensitivity to environmental stimuli and excessive reactions to stimuli. This indicates more intense difficulties in emotional bias in younger athletes. Emotional bias describes asymmetric processing of emotional stimuli by humans, including a negativity bias—meaning an increased response to negative over positive stimuli—and positivity offset. Emotional bias may be modified by different types of stimuli: culture factors, arousal, stimulus type and task setting [27]. We hypothesize that the higher rate of cyclothymic and irritable temperament in young athletes may be related to the level of stress hormones, such as cortisol. However, this requires further research. On the other hand, these findings may indicate a lower ability to manage stress, greater sensitivity to stress stimuli and ineffective coping strategies in younger athletes. Cyclothymic temperament is related to poorer ability to maintain emotional balance in stressful situations, which are common in competitive sports [28]. Individuals with higher cyclothymic traits are also at greater risk of adverse long-term effects of stress, such as depressive and burnout symptoms [29,30]. Finally, the quality of stress coping is related to the level of sport achievement [31]. Higher level of cyclothymic temperament may thus result in poorer coping with stress, leading to adverse psychological effects and poorer athletic performance. This hypothesis warrants further research. Considering the multidirectional relationships between temperament, physical activity and mental health in further studies is important, as exercise is considered as an important factor in prevention of mental diseases [32]. The above observations also shed a new light on athlete training programs, in which education in stress management and the formation of effective coping strategies should be included, especially in young persons. These finding may be of interest for physicians, as they may broaden their knowledge on possible psychological difficulties in young athletes. Awareness of such problems may play an important role in diagnosis and prevention of stress-related diseases in this group.

The main limitations of the present study are related to the size of the study sample, which could result in the occurrence of type II error. However, since this was intended as a feasibility study, definitive conclusions must await larger-scale, adequately powered studies. Another limitation is the lack of information on psychiatric conditions in the study sample, especially on the occurrence of mood disorders which could impact the study results.

## 5. Conclusions

In our study, the TEMPS-A scale of affective temperament appeared to be a useful tool in assessing psychological traits in athletes of different disciplines (aerobic, anaerobic, technical). Studies using this instrument could help to find factors which determine sport achievements in professional athletes. This is important, because we found that the type of affective temperament may be a factor predisposing to involvement in physical activity, though this remains to be confirmed in a larger scale study. This study suggests that the hyperthymic temperament predisposes to physical activity and that depressive and anxious temperaments discourage regular physical activity. Younger subjects appeared to present a stronger tendency to mood lability, higher sensitivity to environmental stimuli and excessive reactions to stimuli. This may indicate more intense problems in emotional bias. More detailed and broader studies are needed to confirm these results, as they may prove to be useful in health promotion and skills training programs dedicated to athletes.

## Figures and Tables

**Figure 1 medicina-58-00195-f001:**
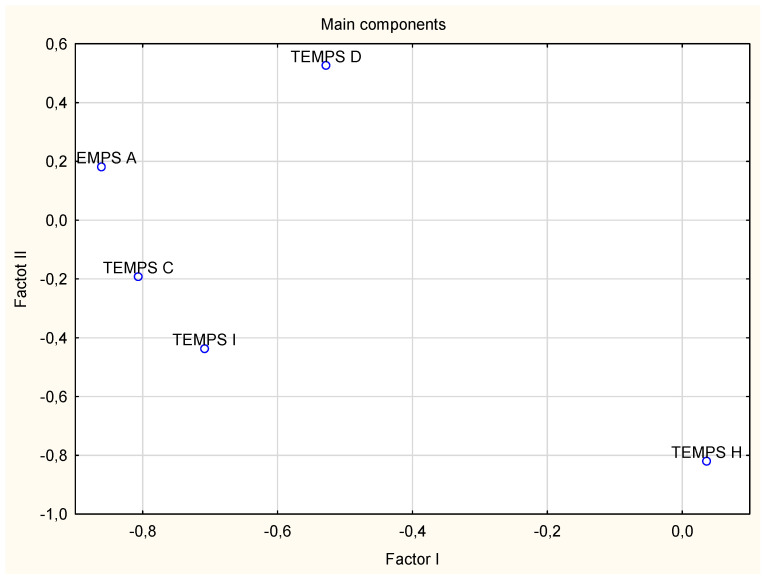
Spatial projection of the principal components of the TEMPS—A subscales.

**Figure 2 medicina-58-00195-f002:**
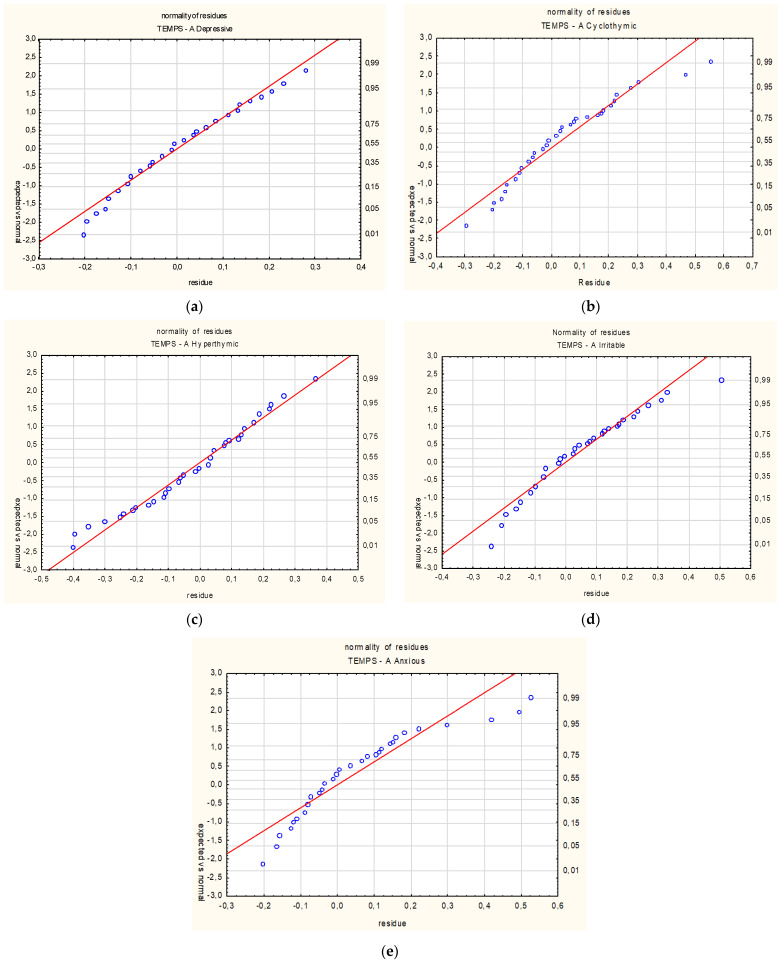
The diagram of normality of residues of affective temperaments dimensions: (**a**) depressive, (**b**) cyclothymic, (**c**) hyperthymic, (**d**) irritable and (**e**) anxious.

**Table 1 medicina-58-00195-t001:** Internal correlation between affective temperaments of TEMPS—A in the study group *N* = 71.

	Depressive	Cyclothymic	Hyperthymic	Irritable	Anxious
Depressive	1.00	0.21	−0.09	0.10	0.49 *
Cyclothymic		1.00	−0.02	0.54 *	0.47 *
Hyperthymic			1.00	0.15	−0.18
Irritable				1.00	0.44 *

* *p* < 0.05.

**Table 2 medicina-58-00195-t002:** Factor analysis of TEMPS—A dimensions.

	Factor	
	I	II
Depressive	0.53	0.53
Cyclothymic	0.81 *	0.19
Hyperthymic	0.04	0.82 *
Irritable	0.71 *	0.44
Anxious	0.86 *	0.18

* Alpha > 0.07.

**Table 3 medicina-58-00195-t003:** The results of TEMPS—A in whole group.

	Whole Group *N* = 71	Females *N* = 33	Males *N* = 38	Differences between Males and Females *p*=
Depressive	0.22 ± 0.11	0.19 ± 0.10	0.25 ± 0.12	0.08
Cyclothymic	0.23 ± 0.17	0.24 ± 0.16	0.21 ± 0.18	0.20
Hyperthymic	0.63 ± 0.15	0.67 ± 0.16	0.60 ± 0.15	0.04
Irritable	0.19 ± 0.15	0.24 ± 0.17	0.16 ± 0.13	0.04
Anxious	0.16 ± 0.15	0.14 ± 0.14	0.18 ± 0.15	0.24

**Table 4 medicina-58-00195-t004:** The results of TEMPS—A in three groups of athletes studied. Mean values ± SD, differences between groups.

	Groups of Athletes			Differences between Three Groups *p* Value	Differences between Groups, Mann–Whitney U Test *p* Value		
	Group I Canoeists	Group II Non-Professional Athletes	Group III Pilots	ANOVA Kruskal Wallis	I vs. II	I vs. III	II vs. III
Depressive	0.20 ± 0.10	0.24 ± 0.14	0.22 ± 0.10	0.74	0.52	0.74	0.78
Cyclothymic	0.30 ± 0.18	0.21 ± 0.17	0.17 ± 0.13	0.01	0.05	0.01	0.53
Hyperthymic	0.63 ± 0.16	0.64 ± 0.18	0.62 ± 0.11	0.58	0.91	0.58	0.57
Irritable	0.24 ± 0.16	0.21 ± 0.17	0.12 ± 0.10	0.01	0.48	0.01	0.03
Anxious	0.16 ± 0.16	0.20 ± 0.18	0.11 ± 0.09	0.29	0.34	0.29	0.05

**Table 5 medicina-58-00195-t005:** Multidimensional analysis.

	Multidimensional Significance Tests. Parameterization with Sigma Constraints. Decomposition of Active Hypotheses					
Effect	test	value	F	df effect	df error	*p*
Free value	Wilks	0.039	313.52	5	64	0.0000
Type of athletes training	Wilks	0.744	2.04	10	128	0.0343

## Data Availability

The data presented in this study are available on request from the corresponding author.

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
