# Peer review of "A Feasibility Study of the Usefulness of the TEMPS-A Scale in Assessing Affective Temperament in Athletes"

_medicina, 2022, doi:10.3390/medicina58020195_

Round 1

Reviewer 1 Report

Thank you for this revised version of the paper. 
All my queries have been addressed.

Reviewer 2 Report

I accept in current form. 

Reviewer 3 Report

The article has been improved according to the reviewers comments and is much more interesting for readers. The current title reflects its content to a greater extent. The authors made a linguistic correction, which significantly improved the quality of the article. The content of the article brings interesting data on the new areas of application of the TEMPS –A questionnaire. In my opinion, the article in its current form can be published in Medicina.

This manuscript is a resubmission of an earlier submission. The following is a list of the peer review reports and author responses from that submission.

Round 1

Reviewer 1 Report

The paper under review presents a cross-sectional study which examines five dimensions of affective temperaments in professional and non-professional athletes. In general, the paper is well written, the presented study is interesting and fills the existing research gap.  The results provide an advance in current knowledge.  Hagop Akiskal's concept of affective temperament is currently used in many studies not only in affective diseases, but also in healthy subjects. The results of various studies on the relationship of affective temperament with stress resistance and predisposition to perform specific tasks are particularly interesting. Moreover, given the rising prevalence of obesity in highly developed countries, the examination of factors related to the level of physical activity is highly relevant. The results of this research indicate a relationship between the dimensions of affective temperament and sports performance. In my opinion, it is interesting to observe that some dimensions of temperament may increase the risk of emotional instability, especially in young athletes. The article uses adequate research methods (authors used the Polish version of the TEMPS-A questionnaire) and the correct statistical methodology.  I recommend the proof-reading of the manuscript, as there are some minor language inaccuracies, for example in the discussion the Authors write:   „results correspond with observations in a bank manager, a paramedic and an athlete” - the names of the paramedic professions should be written in plural.

Author Response

Response to reviewer 1

Thank you for your valuable comments, which will also be used in future studies of affective temperament in various groups of athletes

  1. In this case, term “paramedic” refers to the profession of “paramedic” and does not define a group of medical professions.

Reviewer 2 Report

The submitted manuscript contains many serious errors mistakes in the following areas:
1/  A small group of 71 respondents (questionnaire surveys), 33 men and 30 women.
2/ Separate groups differ in terms of age, which  changes the interpretation of the results significantly, especially in terms of affective temperament.
3/ The analysis group of the 'professional athletes' for sport pilots cannot include athletes between the age of 19-57 due to the experience level withing the group.
4/ In the comparative assessment of professional athletes and non-professional athletes, gender was not taken into account, and the following description is not confirmed by the presented results:
Moreover, female athletes presented a different pattern of temperament profile in comparison with the general populations described in various studies”
5/ The size of the sample and the lack of a strength analysis of the effect call into question the power of the test.

Author Response

Response to reviewer 2

Thank you for your valuable comments, which will also be used in future studies of affective temperament in various groups of athletes

  1. Due to the relatively small group of subjects, the results should be interpreted with caution. Limitations as well as additional clinical implications, were provided in the discussion.
  2. The whole study group was too small to compare the performance of women and men in subgroups of athletes. Only the results of all women and men studied were compared (table 3), and the differences between male and female results were found.
  3. Sport pilots group involve subjects between 23-57 years (mean age 29,8 + 10,2). 18-year-old athletes belonged to a group of canoeists.
  4. The sentence in discussion has been rephrased to reflect the result obtained.
  5. This has been indicated in the work limits, however, the analysis of the TEMPS-A test show its usefulness in the study of athletes. This study is a pilot study, and further studies on a large group of athletes would provide more information.

Reviewer 3 Report

Despite the interesting subject of the research, the paper is written carelessly and requires many corrections and additions.  First, introduction does not provide an interestingly written text, it does not provide a basis for further research. The research material is described briefly. No study criteria. The authors did not present the characteristics of the study group and their subgroups. It is difficult to analyse the results, if there is no good description of the research material. The remaining results are graphically correctly presented, but there is no detailed description of results in the text . The discussion requires significant improvement, the authors briefly referred to the current research in the scientific literature. No test limitations - needs to be completed.

Author Response

Response to reviewer 3

Thank you for your valuable comments, which will also be used in future studies of affective temperament in various groups of athletes

  1. The introduction was provided with the identification of the research gap and the purpose of the research was clarified.
  2. Characteristics of the study group have been provided in the Materials and Methods section
  3. The study results were supported with additional details – results of the Mann-Whitney test, as well as the correlation coefficients, where applicable.
  4. Limitations as well as additional clinical implications, were provided in the discussion.

Reviewer 4 Report

This is an interesting research paper aiming to evaluate the prevalence of different temperament traits in athletes.

Although the topic is innovative and original, I think that there are some aspects to be improved:

  1. at the end of the introduction, authors should better clarify study's hypothesis. Which temperament do you predict to be more frequent? Why?
  2. Is it the first study aiming to evaluate the role of temperament in sport athletes? 
  3. In the methods' section, authors should provide some information related to the study conduction (e.g., assessment tools, inclusion criteria, etc. ). Information regarding the sample characteristics should be moved to the results' section.
  4. In the final part of the discussion, clinical implications of the current study should be specified.
  5. authors should quote some relevant papers on the importance of physical activity for mental health in the general population as well as in people with severe mental disorders (e.g., Firth J, Solmi M, Wootton RE, Vancampfort D, Schuch FB, Hoare E, Gilbody S, Torous J, Teasdale SB, Jackson SE, Smith L, Eaton M, Jacka FN, Veronese N, Marx W, Ashdown-Franks G, Siskind D, Sarris J, Rosenbaum S, Carvalho AF, Stubbs B. A meta-review of "lifestyle psychiatry": the role of exercise, smoking, diet and sleep in the prevention and treatment of mental disorders. World Psychiatry. 2020 Oct;19(3):360-380.
  6. Among socio-demographic information, have you collected any data regarding the presence of any previous mental disorders in participants (or any family history positive for mental disorders)? If not, this should be acknowledged among study's limitations.

Author Response

Response to reviewer 4

Thank you for your valuable comments, which will also be used in future studies of affective temperament in various groups of athletes

  1. The study hypothesis has been clarified and specific temperaments with supposed relationships with athletic performance, were indicated.
  2. This is the first study comparing temperament of professional and non-professional athletes, which was clarified in the introduction. Moreover, previous studies of temperament in athletes, were cited.
  3. Information regarding the sample characteristics has been moved to the results' section.
  4. Clinical implications of the study have been be specified in the final part of the discussion.
  5. The indicated paper was cited in the final part of the discussion.
  6. No such data were collected, and this was outlined in the description of study’s limitations.

Round 2

Reviewer 2 Report

The submitted work contains methodological errors, which are mainly related to the appropriate selection of the study groups. A pilot study is also a research study,

2/ The assessment of affective temperament showed significant differences between men and women. The authors further analyze the affective temperament in sports groups without taking into account gender, despite the fact that they previously indicated that gender is a significant predictor of this temperament. The conclusions of such an analysis are not credible.

 3/ Despite the imprecise age of the pilots, it is still not possible to make a comparative assessment of the affective temperament of pilots aged 23-57, unless the analysis would take into account the professional experience of the studied pilots.

Reviewer 3 Report

Accept in present form.